# The Effect of a Comprehensive, Interdisciplinary Medication Review on Quality of Life and Medication Use in Community Dwelling Older People with Polypharmacy

**DOI:** 10.3390/jcm10040600

**Published:** 2021-02-05

**Authors:** Donna Bosch-Lenders, Jesse Jansen, Henri E. J. H. (Jelle) Stoffers, Bjorn Winkens, Karin Aretz, Mascha Twellaar, Jos M. G. A. Schols, Paul-Hugo M. van der Kuy, J. André Knottnerus, Marjan van den Akker

**Affiliations:** 1Department of Family Medicine, Care and Public Health Research Institute (CAHPRI), Maastricht University, 6200 MD Maastricht, The Netherlands; donna.lenders@maastrichtuniversity.nl (D.B.-L.); jesse.jansen@maastrichtuniversity.nl (J.J.); jelle.stoffers@maastrichtuniversity.nl (H.E.J.H.S.); mascha.twellaar@maastrichtuniversity.nl (M.T.); andre.knottnerus@maastrichtuniversity.nl (J.A.K.); 2Department of Methodology and Statistics, Care and Public Health Research Institute (CAHPRI), Maastricht University, 6200 MD Maastricht, The Netherlands; bjorn.winkens@maastrichtuniversity.nl; 3MEMIC, Center for Data and Information Management, Faculty of Health, Medicine and Life Sciences, Maastricht University, 6200 MD Maastricht, The Netherlands; Karin.Aretz@maastrichtuniversity.nl; 4Department of Health Services Research, Care and Public Health Research Institute (CAHPRI), Maastricht University, 6200 MD Maastricht, The Netherlands; jos.schols@maastrichtuniversity.nl; 5Department of Hospital Pharmacy, Erasmus MC, University Medical Center Rotterdam, 3015 GD Rotterdam, The Netherlands; h.vanderkuy@erasmusmc.nl; 6Institute of General Practice, Goethe University, 60590 Frankfurt am Main, Germany; 7Academic Centre of General Practice, Department of Public Health and Primary Care, KU Leuven, 3000 Leuven, Belgium

**Keywords:** polypharmacy, primary care, medication review, cluster-randomized controlled trial, general practitioner, pharmacist, patient, home visit

## Abstract

**Background:** We conducted a comprehensive medication review at the patients’ home, using data from electronic patient records, and with input from relevant specialists, general practitioners and pharmacists formulated and implemented recommendations to optimize medication use in patients aged 60+ years with polypharmacy. We evaluated the effect of this medication review on quality of life (QoL) and medication use. **Methods:** Cluster randomized controlled trial (stepped wedge), randomly assigning general practices to one of three consecutive steps. Patients received usual care until the intervention was implemented. Primary outcome was QoL (SF-36 and EQ-5D); secondary outcomes were medication changes, medication adherence and (instrumental) activities of daily living (ADL, iADL) which were measured at baseline, and around 6- and 12-months post intervention. **Results:** Twenty-four general practices included 360 women and 410 men with an average age of 75 years (SD 7.5). A positive effect on SF-36 mental health (estimated mean was stable in the intervention, but decreased in the control condition with −6.1, *p* = 0.009,) was found with a reduced number of medications at follow-up compared to the control condition. No significant effects were found on other QoL subscales, ADL, iADL or medication adherence. **Conclusion:** The medication review prevented decrease of mental health (SF36), with no significant effects on other outcome measures, apart from a reduction in the number of prescribed medications.

## 1. Introduction

Multimorbidity and polypharmacy are common in the older population. In the Netherlands, over 85% of people aged 75 years and older have two or more chronic conditions [1] with similar figures across the Western world. Many of these people use multiple medications; when people take five or more chronic prescribed medicines, it is commonly referred to as polypharmacy [2]. Polypharmacy can be appropriate, but many older people are taking medicines that are no longer needed or are unsafe [3]. As life expectancy decreases, the goals of care focus less on prevention and more on quality of life, and the number of medications that can be considered appropriate are significantly reduced [4]. At the same time the potential harm of taking medicines increases, particularly for those who are frail and have multi-morbidity and polypharmacy.

There is extensive literature on polypharmacy showing complex and often conflicting effects on clinical outcomes due to variability in definitions of polypharmacy and outcome measures used with the strongest evidence for hospitalization and inappropriate prescribing [5]. Qualitative work suggests that polypharmacy poses a substantial treatment burden and affects quality of life and adherence [6]. 

Usually, several health care professionals are involved in delivering care for community dwelling older persons, with two-thirds of people with a chronic condition using at least two different health providers (e.g., general practice care, physical therapy, outpatient care, hospital admission). In these situations, well-integrated and coordinated care is of utmost importance to prevent adverse effects and inefficiency. General practitioners (GPs) are ideally placed to strengthen coherence and coordination between primary and secondary care and to help optimize medication management in older community dwelling people [7]. In fact, two thirds of consultations in general practice involve patients with multimorbidity [8].

A vital component of reducing the treatment burden and unnecessary costs associated with inappropriate polypharmacy is to evaluate the evidence on potential benefits and harms of medication in light of the individual older person’s values and overall health and goals. Even if medications are not causing harm, they could be considered inappropriate if they do not align with the older patient’s goals of treatment, their preferences and values [9]. Interventions to reduce inappropriate polypharmacy therefore need to consider the patient perspective and their context [10].

We developed and evaluated an intervention to support medication management of patients with polypharmacy in primary care, the “Polypharmacy Intervention Limburg” (PIL) study. The aim of this intervention was to optimize medication use and to promote appropriate prescribing. Appropriate prescribing in this context is use of medication that is evidence based as much as possible and concordant with patient goals and preferences. It therefore includes well-informed decisions to start or stop medicines. 

This study evaluates the effectiveness of this multidisciplinary patient-centred medication review for patients aged 60 years and older with polypharmacy in primary care on quality of life (primary outcome), functional outcomes and medication changes (secondary outcomes).

## 2. Methods

### 2.1. Design

We conducted a two-armed cluster randomized controlled trial using a stepped wedge design, with unexposed (usual care) and exposed (intervention) observations and a mixed methods process evaluation [11,12]. In the current study, general practices were the level of randomization. Groups of general practices subsequently started the implementation of the intervention in three steps, with three months between each step. Patients received usual care from their GP until entering the intervention. Data collection took place at each of the three steps and at the end of the intervention, and during follow-up (6 and 12 months). See Figure 1 for a schematic illustration of the stepped wedge design.

### 2.2. Sample and Recruitment

*Health Care Professionals.* All general practices in Southern Limburg, the Netherlands, were asked to participate with the PIL study. Twenty-four general practices (43 GPs and 21 practice nurses) agreed to take part in this study, including eight single practices, eight duo practices, and eight medical centers. In addition, seventeen community pharmacies associated with the participating practices agreed to participate. All relevant medical specialists of the two main hospitals in the region (Maastricht University Medical Centre and Zuyderland Medical Centre Heerlen) were informed about the intervention. 

*Patients.* Pharmacists identified patients aged 60 years or over with polypharmacy at the start of the trial. In line with the international literature [2], polypharmacy was defined as chronic use—i.e., more than three months a year—of five or more prescribed medications according to the pharmacy’s information system. GPs reviewed patients’ medical record to identify ineligible patients based on the following criteria: (a) estimated life expectancy of < one year; (b) incompetence to make their own decisions as estimated by the GP and (c) inability to read or write Dutch.

### 2.3. Randomisation and Masking

Randomization of the order in which the GPs received the intervention took place at general practice level in three groups to avoid within-practice, inter-patient contamination.

*Practice level.* A web-based randomization program was used to randomize each of the 24 practices to one of the three groups.

*Patient level.* Each general practice was asked to include 30 to 60 eligible patients. Practices with up to 2350 (1 FTE GP) vs. 2350–3525 (1–1.5 FTEs GP) vs. more than 3525 patients (2 FTE GPs) were asked to recruit around 30 patients, 40 patients and 60 patients, respectively. For each practice, a permutated list was created using an online tool, to determine the order in which eligible patients were invited to participate. Using this list, a trained practice nurse contacted patients by phone providing them with information about the project. Interested patients received a letter with additional information and an informed consent form.

### 2.4. Control Period

Prior to the intervention, patients in the control condition received care as usual. Depending on the group (step), the control period varied between one and six months.

### 2.5. Intervention

Before the intervention, each group (“step”) of GPs and pharmacists attended a training session. In a workshop, moderated by a GP and a pharmacist, they practiced the stepwise PIL medication review. Likewise, a study team GP trained practice nurses in making an inventory of the patient’s medication use at home (for training details, see Appendix A). At the start of the intervention period, a medication review was conducted for each patient, following a systematic multidisciplinary approach comprising six steps. Figure 2 shows the steps of this approach in detail.

### 2.6. Follow-Up Period

The patient entered the follow-up period once all medication changes had been finalized, with the GP reviewing medication use again at intermediate follow-up (around 6 months post intervention) and follow-up at study completion (around 12 months post intervention), to see if further changes were required. At the end of the follow-up period (after 12–16 months), the pharmacist provided a list of all medications dispensed to the patient in the past year to measure patient adherence.

### 2.7. Data Collection

Data collection took place at baseline, before the first group received the intervention, at the start of the intervention, at the end of the intervention and intermediate follow-up and follow-up at study completion. The effects of the trial were evaluated using various patient-related outcomes derived from three sources: the general practices, the pharmacies and the patients (during home visits by the practice nurses; using questionnaires sent at baseline and every three months during the study). See Table 1.

*Background characteristics.* Patients provided information about sex; age (60–69, 70–79, ≥80 years); educational level (none, low, middle, high); living situation (“living independently alone”, “living independently with a partner” or “living in a retirement home”) in written surveys.

*Primary outcome* was quality of life measured using the SF-36 [13] and EQ-5D [14]. The SF-36 consists of eight scaled scores, which are the weighted sums of the questions in their section. Each scale is directly transformed into a 0–100 scale on the assumption that each question carries equal weight. The higher the score the better the quality of life. The eight sections are: vitality, physical functioning, bodily pain, general health perceptions, physical role functioning, emotional role functioning, social role functioning and mental health.

The EQ-5D has two components: health state description and evaluation. In this study we only used the description component to measure quality of life where participants were asked to rate their quality of life on a scale from 0 to 10, with higher scores representing a better quality of life.

### 2.8. Secondary Outcomes

*Daily functioning.* Patient’s functional status was measured using the Activities of Daily Living Scale [15]. Functional limitations were assessed as the required assistance for six basic activities of daily living (ADL) (i.e., bathing, dressing, eating, toileting, use of incontinence products, getting up from a chair) and seven instrumental activities of daily living (iADL) (i.e., grooming, use of telephone, travelling, grocery shopping, meal preparation, household tasks, taking medications, financial management). Responses are scored binary (dependent = 1; independent = 0) and summated, with higher scores representing greater functional limitations.

*Medication use*. The number of medications were extracted from the GP’s and pharmacist’s patient record at the start of the intervention and follow-up; in addition, the number of medications provided by patients during the home visits at the start intervention and through self-report during a home visit at the one year follow-up. A distinction was made between prescribed and over the counter medication. Medication was categorized as “Cardiovascular”, “Diabetes mellitus”, “Digestive tract”, “Lung diseases”, “Psychotropic drugs”, “Analgesics” and “Other medicines”. These categories were the result of previous discussions on medication review by regional groups of GPs affiliated with the Department of Family Medicine of Maastricht University.

*Medication changes made*. Any changes made to medication were compared to the medication list of the previous assessment and were based on patient reported medication (start intervention) and GP data for all other measurements. The concordance between medication prescription (based on GP data), medication dispensing (based on pharmacist data) and actual use as reported by the patient was only 60%; we therefore used patient reports on actual use to assess changes made. This was categorized as no changes, medication stopped, medication changed, reduction in dose and changes in administration. In addition, all medication was categorized by a GP-researcher and assistant, using ATC codes [16].

*Medication adherence.* Medication adherence was measured in the year prior to the intervention and at the one year follow-up, through the dispensing records of the participating pharmacists. By comparing these prescription refill data with the amount of medication prescribed, we calculated a rough adherence estimate. Adherence was measured for chronic medication (≥3 prescriptions a year) only. Excluded were medications to use “when needed” and those dispensed through blister packs (≥20 prescriptions/year).

### 2.9. Analysis

The stepped-wedge design, in which the outcomes are repeatedly measured over time and patients are nested within GP practices, requires statistical techniques that account for the dependency between measurements from the same subject and the same GP practice. To this end, linear mixed-effects model analyses are applied to analyze the data. Briefly, this analysis method addresses the modelling of the multilevel data structure, taking into account the dependence between observations from the same subject and within the same GP practices. The fixed part of the model includes not only group (control or intervention), time (continuous; time^2^ if necessary), and the interaction between group and time (and group*time^2^ if necessary), but also important factors related to the outcome, such as sex, education level, living situation and age. As for the random part, a random intercept on GP practice level (accounting for the nesting of patients within GP practice) as well as random intercept and/or random slope (variance components or unstructured) on patient level (to account for repeated measures) are considered. The final random part is determined using Akaike’s information criterion (AIC). No missing data is imputed as a likelihood-based approach is used to deal with missing outcome data. All analyses were performed using IBM SPSS Statistics for Windows (version 26.0, Armonk, NY, USA, IBM Corp). A two-sided *p*-value ≤ 0.05 is considered statistically significant.

For descriptive analysis data from baseline, start of the intervention and follow-up after approximately 6 and 12 months were used. For the linear mixed models, data were used until intermediate follow-up (around six months), because control time was limited to six months for most participants.

*Sample size.* Due to the multilevel and time-dependent structure of the data with several unknown parameters (e.g., the random part of the model), a valid power analysis was not feasible. However, our study population of 770 participants is substantial compared to other polypharmacy trials in primary care; the systematic review of Rankin et al. [17] included 10 primary care polypharmacy trials with 99–516 participants, with a mean of 259.

From October 2010 to February 2012 we collected data from 774 eligible patients aged 60 years or older with polypharmacy. We have included 770 patients. Of these participants, 134 (17.4%) were lost to follow-up for various reasons (43 died, 32 withdrew participation, 18 changed GP, 5 moved to a nursing home, 4 people’s health deteriorated, 1 moved to a hospice, 31 unknown reasons).

### 2.10. Baseline Characteristics

Participants included 360 female and 410 male patients with an average age of 75 years (SD 7.5). The majority was born in the Netherlands, had a lower educational background and lived independently (See Table 2).

### 2.11. Data are Number of Participants (%), Unless Otherwise Stated 2.10. Primary Outcome

In terms of quality of life measured with the SF-36, mental health decreased from baseline to start of intervention to end of study (see Table 3). Table 4 shows that there was a statistically significant difference in trend over time between the control and intervention condition (*p* = 0.009): mental health (estimated mean) was stable in the intervention group, but decreased in the control condition from 80.9 at baseline to 74.8 after six months, i.e., average of 1.0 points per month. No significant different changes were observed on the other SF-36 subscales or on the EQ-5D.

### 2.12. Secondary Outcomes

There were no statistically significant differences in changes over time in (instrumental) activities of daily living across the different time points (*p* = 0.608). Time trends for ADL and iADL were not different for the intervention and the control condition (Table 4) Median medication adherence based on dispensing records of the participating pharmacists was high at both baseline (97.7 out of 100) and follow-up at the end of the study (98.2 out of 100) (Table 5). Overall, patients were prescribed fewer medication at the end of the study (total medication 4318; mean 7.2) compared to start intervention (total medication 5527; mean 7.6) (Table 6).

## 3. Main Results

In this primary care study among 770 patients aged 60+ years with polypharmacy, we explored medication changes and medication adherence and tested the effect of a comprehensive medication review on patients’ quality of life and (instrumental) activities of daily living (ADL and iADL). Results showed that during the intervention period, changes were made to medication (medication stopped, dose reduced but also added). Moreover, the medication review resulted in a positive effect on mental health over a period of six months (measured with SF36). This seems to be due to stable mental health during the exposed period (intervention), whereas participants’ mental health declined during the unexposed period (usual care). No changes were found for the other quality of life scales: ADL, instrumental ADL, and medication adherence was high at baseline and follow-up.

### 3.1. Strengths and Limitations

Some methodological considerations should be considered. First, selection of patients might have occurred, since the patients in our study reported high quality of life (e.g., SF36 QoL General Health at baseline in our study 53.4; compared with scores ranging from 56.9 to 44.4 in older adults aged 65–69 years and 85+ years, respectively [18]) and good functional status (e.g., an average ADL score of 0.49 out of 6 at baseline in our study compared with 1 out of 6 in a sample of community living older adults with a mean age of 80 years [19]) at the start of the intervention [20], and their medication adherence was good (97.7 out of 100 at baseline). Future research is needed to identify subgroups of patients for whom these types of interventions are of most benefit, for instance the very old, those with frailty/disability, and limited life-expectancy or those with hyper polypharmacy [21] or experiencing a high treatment burden; and to identify the medication and disease clusters with the potential for benefit [5].

Our intervention addressed key known barriers to appropriate prescribing in older patients [22]. For instance, it involved all key stakeholders (GPs, pharmacist, specialist), included elicitation of patient preferences and goals, and recommendations were slowly implemented in collaboration with the patient and careful monitoring by the GP. At the same time however, we cannot draw conclusions about the effectiveness of the specific ingredients of medication reviews.

While we observed changes in the number of medications during the follow-up period, thereby reducing polypharmacy, a limitation is that we could not investigate in detail if these medication changes were appropriate. A recommended core outcome set suggests differentiating between overuse (use or prescription of more drugs than clinically needed), underuse (failure to prescribe drugs that are indicated), potentially inappropriate medications and high-risk medications [23], but patient preferences should also be taken into account when judging medication changes. The follow-up period of one year may also have been too short to identify relevant changes in quality of life and functioning.

A limitation is the lack of blinding of patients and clinicians, which was not feasible due to this design. However, this is unlikely to affect the measurement of objective outcomes such as the number of medications used.

### 3.2. Comparison to Previous Studies

A common criticism of medication review trials is that the outcome measures used are too broad with many influencing factors, which would make it difficult to show the effect on an intervention that is specific to medication only [5,24]. However, we felt that quality of life was the most appropriate primary outcome measure in this study given the comprehensive nature of the intervention. This is supported by a recent trial in which a medication review improved quality of life measured by the EQ-VAS (but not EQ-5D-5L) [25]. Moreover, quality of life was identified as a core outcome of medication review trials in a recent international consensus study [23]. Other patient-related outcomes may include medication or treatment satisfaction, which may reflect the successful application of shared decision making in medication prescription or deprescribing.

While previous studies suggested that polypharmacy reduces adherence [26], this was not confirmed by our findings as adherence was very high both at baseline and at follow-up. These findings are in line with a growing body of work showing that medication reviews and other interventions to reduce inappropriate polypharmacy have an effect on medication-related outcomes such as decrease in the number of medication related problems, medication dosage and number of medications prescribed (or a smaller increase) but with mixed findings related to quality of life or other patient outcomes [5,24]. The adherence measure in our study was measured using the possession of medication as a proxy and not actual intake, it therefore is likely to result in an overestimation of actual adherence [23].

## 4. Conclusions

Descriptive analysis showed that our comprehensive medication review resulted in medication changes, with a positive effect on mental health (as measured with SF36). The intervention did not impact on other patient outcomes including quality of life measured with EQ-5D, (I-)ADL; and medication adherence was high at baseline and follow-up. All in all, our results show that the medication review did not decrease patients’ QoL, which could be considered a positive finding.

## Figures and Tables

**Figure 1 jcm-10-00600-f001:**
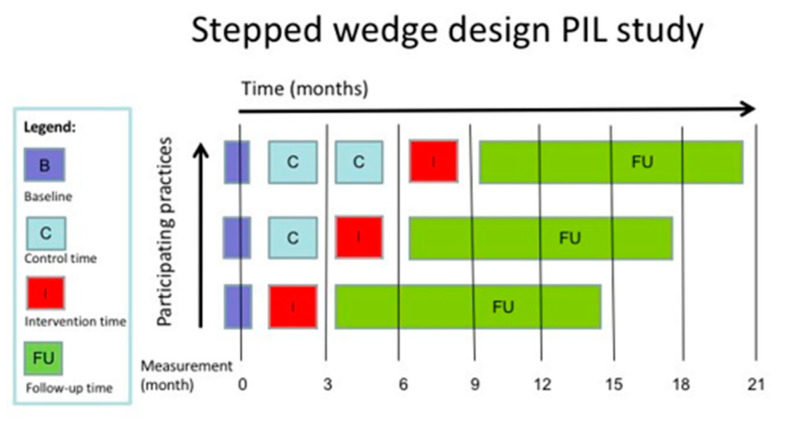
Flowchart stepped wedge design.

**Figure 2 jcm-10-00600-f002:**
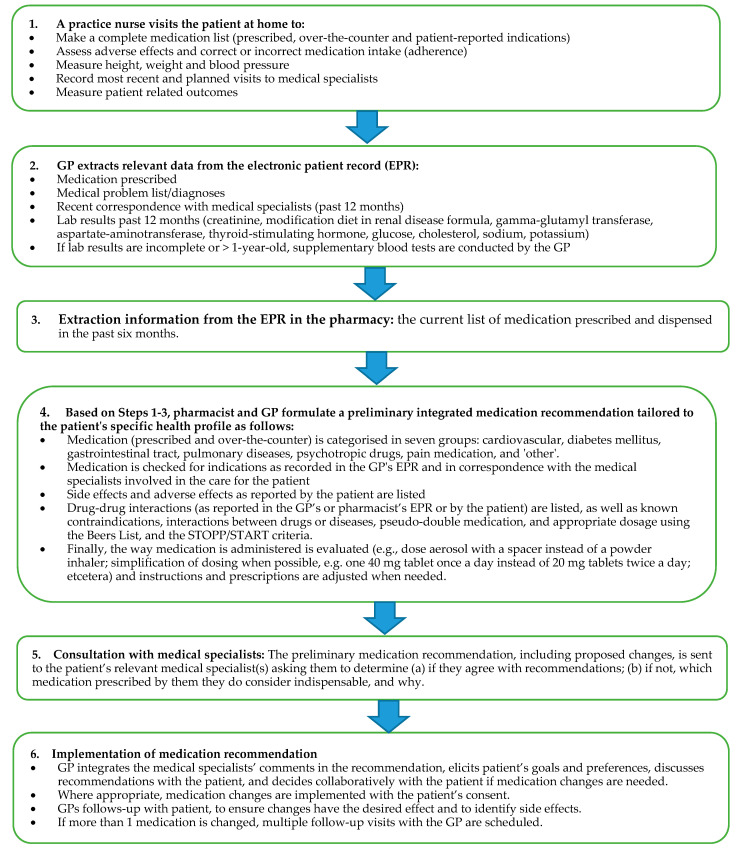
The Intervention: Integral Medication Control and Monitoring System.

**Table 1 jcm-10-00600-t001:** Data collection: concepts measured at General Practice (GP), Pharmacy, and Patient for the Polypharmacy Intervention Limburg (PIL) trial.

	Baseline	Intervention		Follow-Up	
		Start	End	Intermediate	End of Study
*Data provided by GPs (GP’s EPR *)*					
Current medication list		X	X	X	X
Laboratory results ^1^		X			
Diagnoses (ICPC-1)		X			X
Correspondence with specialist(s) involved		X			
*Data provided by pharmacists (pharmacy’s EPR)*					
Current medication list		X		X	X
Medication delivery details and adherence		X		X	X
*Data provided by patients*					
*Home visits*					
Current medication use reported by patient		X			X
Use of over-the-counter medication		X			X
Adverse drug effects		X			X
Knowledge of medication use		X			X
Appropriate use of prescribed medication		X			X
Correct storage and use of medication		X			X
Use of medication-related devices/assistance		X			X
Use of food/drinks that can influence health status		X			X
Mobility		X			X
Stool and micturition habits		X			X
Height		X			X
Weight		X			X
*Patient questionnaires (by mail)*					
Quality of life (SF-36, EQ-5D)	X	X	X	X	X
Activities of daily living (ADL-I-ADL)	X	X	X	X	X
Date and country of birth, gender, education	X				
Postal code, living situation	X	X	X	X	X

^1^ Creatinine, MDRD, GGT, ALAT, glucose, sodium, potassium, urea, TSH, FT4, hemoglobin, hematocrit, Mean Cellular Volume. * EPR = electronic patient record.

**Table 2 jcm-10-00600-t002:** Background characteristics of patients at baseline (*N* = 770).

*Age*	
60–69	304 (39.6)
70–79	300 (39.0)
≥80	166 (21.6)
*Sex*	
Female	359
Male	409
Missing	2
*Country of birth*	
The Netherlands	722 (94.6)
Other European country	27 (3.5)
Indonesia	11 (1.4)
Other	3 (0.3)
Missing	7
*Level of education*	
None	185 (24.3)
Low	358 (47.1)
Middle	129 (17.0)
High	88 (11.6)
Missing	10
*Living arrangements*	
Independent, with someone	509 (67.1)
Independent, alone	213 (28.1)
Nursing home	37 (4.9)
Missing	11
*Average number of medications/patient*	
Reported by the patient	8.0 sd 2.6
From the GP records	7.4 sd 2.5
From the pharmacist records	7.5 sd 2.4

**Table 3 jcm-10-00600-t003:** Patients’ quality of life (SF-36 and EQ-5D) activities of daily living (ADL) and instrumental ADL, at baseline, start of the intervention, intermediate follow-up and end of study.

Outcome	Baseline (*N* = 768) ‡			Intervention			Follow-Up ¥					
				Start (*N* = 746)			Intermediate (*N* = 624)			End of Study (*N* = 508)		
*Quality of life: SF-36* *	**Mean**	**SD**	**Median**	**Mean**	**SD**	**Median**	**Mean**	**SD**	**Median**	**Mean**	**SD**	**Median**
Physical functioning	57.9	28.5	60	58.3	29.0	60	58.1	29.3	60	58.4	28.1	60
Social functioning	77.0	25.1	87.5	76.8	24.9	87.5	74.7	25.8	75	75.7	24.1	75
Role physical	54.4	44.4	66.7	54.4	44.8	75	53.7	44.9	75	50.5	44.4	50
Role emotional	77.0	37.7	100	75.8	38.0	100	73.6	38.9	100	74.0	39.7	100
Mental Health	76.7	17.7	80	76.4	18.1	80	75.6	17.8	80	74.8	18.1	80
Vitality	61.8	19.8	65	62.0	20.0	65	61.2	19.5	65	60.0	19.1	60
Pain	66.0	25.7	67.3	66.2	26.1	67.3	64.6	26.2	67.3	64.7	24.4	67.3
General health	53.4	18.3	55	54.1	18.5	55	52.9	18.1	50	52.7	17.9	50
*Quality of life(EQ-5D)* ^$^	7.6	2.1	8.1	7.7	2.1	8.1	7.6	2.2	8.1	7.6	2.0	8.1
*Activities of daily living (ADL/iADL)* ^#^												
ADL	0.49	0.91	0	0.50	0.90	0	0.54	0.97	0.	0.52	0.92	0
Instrumental ADL	1.01	1.43	0	1.05	1.51	0	1.01	1.40	0	1.01	1.49	0

‡ Numbers vary due to missing values. ¥ Intermediate follow-up after 6–9 months; End of study after 12–16 months. * Total scores can range from 0–100: higher scores indicate better quality of life. $ Total scores can range from 0–10: higher scores indicate better quality of life. # Total scores can range from 0–6 for ADL and 0–7 for iADL, representing the number of activities that require assistance.

**Table 4 jcm-10-00600-t004:** Effects of a comprehensive medication review in patients with polypharmacy in primary care; results of a Linear Mixed Model Analysis of patient outcomes comparing control and intervention at start of period and after six months (*N* = 746–752) ‡.

Outcome		Start of Period	After 6 Months	Difference in Overall Trend Between Control and Follow-Up; *p*-Value
*Quality of life: SF-36* *		Estimated mean (95%CI)		
Physical functioning	Control: Intervention:	52.3 (47.2, 57.4) 52.8 (47.6, 57.9)	53.8 (48.3, 59.2) 52.0 (46.9, 57.1)	0.057
Social functioning	Control: Intervention:	80.8 (70.3, 91.3) 68.0 (61.6, 74.5)	74.0 (66.9, 81.1) 69.0 (63.1, 75.0)	0.071 (**)
Role physical	Control: Intervention:	45.4 (34.7, 56.0) 43.7 (32.9, 54.4)	44.1 (32.5, 55.6) 42.9 (32.3, 53.5)	0.855
Role emotional	Control: Intervention:	77.0 (67.3, 86.7) 72.5 (62.7, 82.3)	74.4 (63.8, 85.0) 72.5 (62.8, 82.1)	0.327
Mental Health	Control: Intervention:	80.9 (74.5, 87.3) 71.6 (67.5, 75.7)	74.8 (70.3, 79.3) 71.3 (67.5, 75.1)	**0.009** (**)
Vitality	Control: Intervention:	58.0 (54.0, 61.9) 57.9 (53.9, 61.9)	58.1 (53.8, 62.3) 56.8 (52.9, 60.8)	0.246
Pain	Control: Intervention:	71.2 (65.5, 76.9) 71.3 (65.5, 77.0)	70.9 (64.8, 77.1) 70.2 (64.6, 75.9)	0.570
General health	Control: Intervention:	49.9 (46.0, 53.7) 49.3 (45.4, 53.2)	50.2 (46.1, 54.4) 48.8 (45.0, 52.7)	0.373
*Quality of Life * (EQ-5D)*	Control: Intervention:	6.7 (6.2, 7.2) 6.7 (6.2, 7.2)	6.8 (6.3, 7.3) 6.7 (6.2, 7.1)	0.082
*Activities of daily living (ADL)* †				
ADL	Control: Intervention:	0.97 (0.79, 1.16) 0.99 (0.81, 1.17)	0.98 (0.78, 1.17) 1.02 (0.83, 1.12)	0.551
Instrumental ADL	Control: Intervention:	1.82 (1.54, 2.09) 1.84 (1.57, 2.12)	1.87 (1.58, 2.16) 1.86 (1.59, 2.14)	0.608

Note that the means in this Table differ from Table 3 due to the assumptions made by the Linear Mixed Model Analysis. ‡ Numbers vary due to missing values * Total scores can range from 0-100: higher scores indicate better quality of life. † * Total scores can range from 0-10: higher scores indicate a better quality of life. ** Overall test for difference in linear and quadratic trend.

**Table 5 jcm-10-00600-t005:** Analysis of number of medications.

	Start Intervention *N* = 770	End Intervention *N* = 745	Follow-Up #	
			Intermediate *N* = 603	End of Study *N* = 597
Total number of medications *:	5469	5527	4407	4318
Average number of medications *:	7.4 sd 2.5	7.6 sd 2.5	7.3 sd 2.4	7.2 sd 2.5
*Medication according to category*				
Cardiovascular	3030 (4.1 sd 1.8)	3042 (4.1 sd 1.9)	2452 (4.1 sd 1.8)	2411 (4.0 sd 1.8)
Diabetes Mellitus	395 (0.5 sd 0.9)	404 (0.5 sd 0.8)	357 (0.6 sd 0.9)	339 (0.6 sd 0.9)
Digestive tract	447 (0.6 sd 0.7)	451 (0.6 sd 0.7)	372 (0.6 sd 0.7)	351 (0.6 sd 0.9)
Lung diseases	308 (0.4 sd 0.9)	324 (0.4 sd 0.9)	240 (0.4 sd 0.9)	252 (0.4 sd 0.9)
Psychotropic drugs	212 (0.3 sd 0.6)	201 (0.3 sd 0.5)	148 (0.2 sd 0.5)	148 (0.2 sd 0.5)
Analgesics	178 (0.2 sd 0.5)	173 (0.2 sd 0.5)	120 (0.2 sd 0.5)	123 (0.2 sd 0.5)
Other	899 (1.2 sd 1.3)	932 (1.3 sd 1.4)	718 (1.2 sd 1.3)	694 (1.2 sd 1.3)
*Median medication adherence* ** 8 (min-max)	97.7 (74–100)	-		98.2 (33–100)

Data are number of participants (%), unless otherwise stated. # Intermediate follow-up after 6–9 months; End of study after 12–16 months. *As reported by the GP ** Based on dispensing records of the participating pharmacists. Due to skewed distribution the median is reported, total scores can range from 0–100: higher scores indicate better adherence, a score > 90% is considered adherent.

**Table 6 jcm-10-00600-t006:** Analysis of secondary outcomes medication changes.

	End Intervention	Follow-Up	
		6 Months	12 Months
*Changes made to medication*			
None	4473 (70.7%)	3367 (69.8%)	3367 (69.8%)
Medication stopped	798 (12.6%)	673 (14.9%)	616 (14.2%)
New medication added	339 (5.4%)	451 (8.9%)	437 (8.9%)
Dose changed	482 (7.6%)	305 (6.7%)	247 (5.6%)
Restarted	237 (3.7%)	103 (2.0%)	158 (3.2%)

Data are number of participants (%), unless otherwise stated.

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
