# Peer review of "The Effect of a Comprehensive, Interdisciplinary Medication Review on Quality of Life and Medication Use in Community Dwelling Older People with Polypharmacy"

_jcm, 2021, doi:10.3390/jcm10040600_

Round 1
Reviewer 1 Report
Polypharmacy affects a large and growing proportion of the population and is a significant burden to patients and the health system alike. Delivery of medication optimisation is a key aspect of clinical practice. The authors address a research question that is highly important and relevant at the current time and where evidence is still lacking. The study design, a stepped wedge cluster randomised controlled trial, is appropriate to answer the underlying research question.
Comments and suggestions
- ‘Methods’, subsection ‘design’ - this article would benefit from further details of the methods used. The stepped wedge design presented in some parts is unclear, inclusion of a schematic illustration of this stepped wedge trial showing; the time points at when each cluster crossed-over into the intervention and the separate observation periods based upon the four data collection points (e.g baseline, end of intervention, intermediate follow-up, end of study) would be helpful to the reader.
- Line 102 – Use of the term ‘two-armed’ is unusual in a stepped wedge trial, exposed and unexposed observation periods may be more appropriate terminology to use here.
- Subsection ‘Sample and recruitment’, line 111 – it is unclear whether all participating were recruited at the start of the trial and followed-up over the 3 steps or some participants joined during the trial and were followed up for the remaining steps. Clarification here would be helpful.
- Subsection ‘Intervention’, line 149 – The comprehensive medication review is well described. In the Discussion section, Line 353- ‘elicitation of patient goals and preferences’ is mentioned however this is not explicit within the intervention description given in the text or in Box 1. Please clarify for the reader how patient goals and preferences were obtained or remove the reference to this in the discussion section.
- Subsection ‘medication use’, lines 209-210 suggest that the current medication list was collected at baseline from the GPs EPR, this data collection is not however reflected in Table 1.
- Line 209 clarification is required as to whether data pertaining to the number of ‘prescriptions’ or ‘medications’ were extracted from the patient record, I think that this sentence should probably include both terms e.g. prescription medications.
- As per point 5 above, subsection ‘medication changes made’, lines 220-221 state that patient reported medication use was captured at baseline, this is not presented in Table 1.
- Subsection ‘medication changes made’, lines 220- 229- this section describes collection and categorisation of data related to changes made to medication use over time. Findings for changes made to medications however are not presented within the results tables or reported in subsection ‘secondary outcomes’. Furthermore, ‘Main results’, line 336 – reports that substantial changes in medication (medication stopped, dose reduced/added) were made because of the medication intervention. Data showing changes in medication over time should be included within the tables of results to corroborate this. Alternatively, subsection ‘medication changes made’ and any reference to changes in medication should be removed from the manuscript.
- Lines 193-196 this sentence would benefit from some revision to clarify that only the health state description component of the EQ-5D was used for measurement of quality of life.
- Table 5. Total number of medications- some data presented here require confirmation, specifically, sd for diabetes medication and % and sd for analgesics
- ‘Main results’ lines 333 and 337- as per point 8 above, findings related to changes made to medication are not corroborated by what is presented in the tables of results or the related text. These sentences should be amended to better reflect the conclusions presented within the abstract.
- ‘Strengths and limitations’ whilst the quality-of-life score, functional living scores and medication adherence of this study population appear high, I do not have a feel for how this relates to these measures in the general population aged 60 years or more. Inclusion of evidence and a reference in this regard to enable comparison between the populations is required.
- Lines 368-371 - The methods and results section report upon medication use only (as per points 8 and 12 above no data related to changes made to medication is presented). Furthermore, medication use data presented in results Table 5 is based upon data captured from the GP EPRs only. As such statements related to the development of a reliable measure of medication use and changes made to medications are currently unsupported.
- ‘Conclusion’, line 408- I agree that the descriptive analysis shows a reduction in the number of medications, however, the suggestion that medication dosage was reduced is currently unsupported. As per point 12 this sentence should be amended to better reflect the conclusion presented in the abstract.
Reviewer 2 Report
The paper describes a cluster-randomised stepped wedge multidisciplinary medication review intervention in primary care. The paper is very well written and adds value to the current literature on medicine optimisation.
Few issue the authors might consider
1- in the method section on page 4-Intervention- the authors mentioned that GPs and pharmacists received training on medication review before starting implementing the intervention. it would be really useful to the reader to understand and learn what the training involved and how was conducted. so more information about the training is useful. if word count does not permit this, I suggest to provide the information as additional files
2- on page 5 please review and rephrase the following "The EQ-5D has two components: health state description and evaluation, in this study we only one descriptive question were participants were asked to rate their quality of life on a scale from 0 to 10, with higher scores representing a better quality of life".
3- Box 1 includes lots of information, I wonder whether it is possible to present the information in a different way such as a flow diagram!
4- in the discussion, the authors failed to really comment and explain why the primary outcome "quality of Life" has not changed between the intervention and the control. they should really give some insight on why they believe this is the case. and also comment on why improvement were seen with the mental health aspect of quality of life! could this be due to the improvement in cognition or function?? or other reasons. adding some explanations or suggestions for this is very important information that need to be added to the paper to strengthen it.
